# Surface Texturing of Cylinder Liners: A Review

**DOI:** 10.3390/ma15238629

**Published:** 2022-12-03

**Authors:** Pawel Pawlus, Waldemar Koszela, Rafal Reizer

**Affiliations:** 1Faculty of Mechanical Engineering and Aeronautics, Rzeszow University of Technology, Powstancow Warszawy 8 Street, 35-959 Rzeszow, Poland; 2Institute of Materials Engineering, College of Natural Sciences, University of Rzeszow, Pigonia Street 1, 35-310 Rzeszow, Poland

**Keywords:** surface texturing, cylinder liner, friction, wear, oil consumption

## Abstract

The effect of cylinder liners on engine performance is substantial. Typically, the cylinder surfaces were plateau honed. However, recently additional dimples or grooves were created on them. This work discusses the tribological impacts of textured cylinder liner surfaces based on a review of the literature. The results of the experimental research obtained using test rigs and fired engines were critically reviewed. In addition, the results of the modeling are shown. Circular oil pockets and grooves perpendicular to the sliding direction of piston rings of small depths were typically used. Surface texturing of the cylinder liners governs lubrication between the cylinder liner and the piston ring by an increase in oil film thickness near the reversal points leading to reductions in friction force and wear and in the fired engine to a decrease in fuel consumption and to an increase in power or torque. The correct texturing pattern ensures a decrease in the oil consumption, blow-by, and emissions of the internal combustion engine compared to plateau-honed surfaces. Considerations of future challenges are also addressed. The volume of lubricant reservoir in surface topography, called oil capacity, should be a substantial parameter characterizing textured surfaces.

## 1. Introduction

Forty-five percent of the frictional losses, on average, in passenger cars [1] and trucks and buses [2] is consumed by the piston assembly. Most of these losses originated from the co-action between piston rings and the cylinder liner. The surface of the cylinder liner co-acts with the surfaces of the piston skirt and piston rings. The piston ring assembly operates in difficult conditions (maximum temperature, maximum gas pressure, and null sliding speed) near the top dead center (TDC); therefore, mixed or boundary lubrication occurred [3]. The friction losses of the piston ring pack can be reduced by surface texturing of the piston ring and/or the cylinder liner. The surfaces of the piston skirts were rarely textured, due to applications of coatings that reduce friction. Surface texturing is an option of surface engineering depending on the creation of dimples (oil pockets or cavities) or valleys on typically one surface in contact. It can also help to reduce abrasive wear and the tendency to seizure [4,5]. There are many reviews related to surface texturing [3,6,7,8,9,10,11,12,13,14]. Among the texturing techniques, laser texturing is the most popular [6], although there are other techniques. Among alternatives to laser texturing, burnishing [4,5,15,16] or abrasive jet machining [17,18] are promising techniques.

The first possibility is to texture the surfaces of the compression (first) piston ring. Its radial surface has an influence on contact with the cylinder wall. Ryk et al. [19] found in experimental research using a test rig that full laser surface texturing led to a reduction in friction force by up to 40%. The introduction of partial laser surface texturing applied symmetrically at both axial ends of the rings caused an additional reduction in friction force by up to 25% [20,21]. Partial laser surface texturing led to a reduction in fuel consumption by internal combustion engines to 4% [22]. Shen and Khonsari [23] and Zhang et al. [24] found that the laser surface texturing of the piston rings led to running-in shortening and improved friction and sealing performances. Oil pockets had trapezoidal [23], and rectangular and circular shapes [24]. Ezhilmaran et al. [25] and Gu et al. [26] revealed that laser surface texturing of piston rings with circular dimples caused reductions in friction. Miao et al. [27] found that dimples on the piston ring surface are better-accumulated lubricant than grooves on the cylinder liner. The dimpled texture of the piston rings led to a better increase in contact resistance than the grooved texture of the cylinder liners [28]. Rao et al. [29] studied the tribological effect of the grooved cylinder liner accompanied by the dimpled piston ring radial surface and found that the first option reduced exhaust gas (Nox) emissions. Partial laser surface texturing of the coated piston ring led to a reduction in friction [30]. More information on the effects of piston ring surface texturing on the tribological performance of the piston ring pack can be found in reviews [31,32,33,34]. Despite some advantages, surface texturing of piston rings was not implemented in mass production yet.

Materials of cylinder liners are selected on the basis of operational conditions, engine performance, and service life. Cast iron is the widely used material for cylinder liners, followed by steel. Due to the tendency to apply lighter materials, cast iron and steel are replaced by aluminum alloys, such as Al-Si. After honing, aluminum bores are coated with wear-resistant coatings such as NIKASIL^®^ coating.

Cylinder liner honing was one of the first types of surface texturing applications. First, one-process honing was applied, and the created cross-hatched texture had an ordinate distribution similar to Gaussian. Then, plateau honing of cylinder liners was introduced. The formed two-process texture [35] should have better properties than one-process honing, such as shorter running-in and lesser wear than a one-process surface of the same average roughness height [36]. Santochi and Vignale [37] obtained a smaller fuel consumption and higher power of engines with the two-process (plateau-honed) cylinder texture compared to the one-process surface. Typically, plateau-honed surfaces led to better tribological performances than the one-process honed cylinder liner surfaces. Pawlus obtained lower wear of two-process texture during running-in [38] and during engine operation under artificially increased dustiness conditions [39]. Grabon et al. obtained a smaller coefficient of friction [40] and wear of plateau-honed cylinder surfaces [41,42]. Yin et al. [43] achieved a lower friction torque after plateau honing. However, the authors of the works [44,45,46,47] did not obtain an improvement in functional properties due to plateau honing. Honing angle is a substantial parameter. In most cases, better engine performances were obtained by decreasing the honing angle, such as a reduction in fuel consumption [48] and an increase in minimum film thickness [49]. The honing angle is typically between 45 and 65°; however, in helical slide honing, an angle of 140–150° can be applied [50]. There is a tendency to decrease the height of the cylinder liner, which leads to a decrease in wear, oil consumption, and engine exhaust emission. Therefore, slide honing with diamond sticks is used [46,51]. Paper [52] reviews the functional performance of honed cylinder liner surfaces. Cylinder liner surfaces after honing and, particularly, plateau honing are difficult to characterize. Therefore, various methods have been developed, for example, to characterize plateauness [46,53].

Dimples on the cylinder liner surface may improve the tribological performance of the piston ring pack. Contrary to piston rings, laser texturing cylinder liners were applied in production. For example, this process was introduced at Opel Powertrain Company in 2002 [54]. Reviews [31,32,33,34] are related to texturing of both the piston rings and the cylinder liner. Only cylinder liners with additional dimples and grooved will be analyzed in this paper. The impact of honing, described in [52] will not be considered in detail. This review will be divided into three parts: the results of tests using laboratory simulators, the results of tests of fired (hot) engines, and the results of modeling.

## 2. Functional Performance of Textured Cylinder Liner Surfaces

### 2.1. Laboratory Simulators

Laboratory simulation tests are often used because of the good availability of tribotesters. Their advantages are good repeatability and the possibility of obtaining direct influence of cylinder liner textured surfaces on tribological properties (typically friction and wear). Typically, the detail of the piston ring co-acts with the detail of the cylinder liner (Figure 1). The other possibility is to use an internal combustion engine powered by an electrical motor. In this case, the conditions become more similar to those existing in a fired engine than in classical test rigs.

When classical rigs are used, the experiments are carried out in lubricated motion. Usually, load, frequency of oscillations, and temperature are test parameters. The stroke is typically smaller than that of the engine. It is good to obtain the course of the friction coefficient within one stroke [55]. Due to the high hardness, wear of textured cylinder liners is typically low; therefore, it can be assessed using the profilometric method [53,56,57,58]. Under repeatable conditions, the effect of surface texture on tribological properties is often substantial. However, it is very difficult to obtain conditions similar to those existing in the real internal combustion engine—this is the disadvantage of research using classical test rigs.

Duffet et al. [59] used Nd:YAG laser beams to create oil pockets on the cylinder liner surface. Due to the existence of holes, the lubricant stayed more in contact leading to longer operation with a small coefficient of friction which then increased (the oil was supplied before the test). During tests, the frequency was 1 Hz, the stroke was 10 mm, and the normal load was 450 N. Creating oil pockets was necessary because in modern engines the oil film thickness is near 1 µm or lower.

Tomanik [60] compared the tribological behavior of the slide-honed cylinder liner with that of the cylinder liner with isolated dimples of oval shape sliding against PVD-coated piston rings in a reciprocating motion. The oil pockets had lengths of 1 and 3 mm. The liner surface after laser texturing with lengths of pockets of 3 mm and a very smooth surface showed the smallest friction. The influence of liner surface texturing on liner wear was marginal. Ring wear decreased for smoother liner surfaces.

Zhu et al. [61] found that the greatest effect on friction reduction had a variable dimple density on the cylinder liner surface—the lower density corresponded to the middle region and was higher at both ends.

Oil pockets of 10 µm in depth and 100 µm in width and with a pit area ratio of 12.5% were made on cast iron surface. Experimental tests were carried out under step-loading conditions using an SRV tribometer. Untextured samples led to seizure contrary to the textured sliding pair [62].

The effect of the dimple array on tribological performance was seldom analyzed. Zhan and Yang [63,64,65] performed some research in this field. In [63] they studied the effect of θ angle (Figure 2) on the wear of cylinder liners under starved lubrication. The diameter of the dimple was 100 µm, the dimple depth was near 25 µm, and the pit area ratio was 20%. The smallest wear was obtained for θ angle equal to 60° (Figure 3). The wear of the dimpled cylinder liner was smaller than that of the honed cylinder liner, approximately 20% less. In [64] the cylinder liner co-acted with the first barrel-shaped ring. The stroke was 6 mm.

For the ratio of dimple depth to diameter of 0.18, pit area ratio of 20% and θ angle of 60° surface texturing of cylinder liner caused a reduction in the friction coefficient of 50%, of cylinder wear of nearly 86%, and of the first piston ring of 50% under full lubrication. The parameters of the dimples were selected based on lubrication theory. However, in starved lubrication, surface texturing caused a reduction in cylinder wear by almost 35% [64]. In [65] the smallest wear of the cylinder was obtained for the angle of 45º in full lubrication (dimple diameter was 0.1 mm, the ratio of the depth of the dimple to diameter was 0.1, and pit area ratio was 20%).

Liu [66] compared the performances of smooth, honed, and honed surfaces with additional dimples using an Optimol tester at a high temperature (400 °C). The stroke was 1 mm, the frequency of oscillation was 20 Hz, and the normal load was 300 N. The smallest wear and friction were obtained for the honed surface. However, in this paper, information on the geometry of the dimple pattern was not provided.

Grabon et al. [67] added dimples created by the burnishing technique to plateau-honed cylinder liner surfaces from gray cast iron. The depth of the circular oil pockets was up to 10 µm, the diameter was between 0.15 and 0.2 mm, and the pit area ratio was 13% (Figure 4). It was found that in good lubrication conditions the friction force was reduced by up to 50% compared to operation with the untextured surface (Figure 5). Additional texturing led to an increase in the oil capacity of 50%. However, in a starved lubrication regime, the beneficial effect of additional liner surface texturing was marginal.

Morris et al. studied the influence of chevron patterns located on the surface of the cylinder liner on friction reduction in reciprocating sliding [68] (Figure 6). The depth of the chevrons was 3 µm. They analyzed also results obtained by Costa and Hutchings [69].

Vlădescu et al. [70] found that grooves of nearly 8 µm depth normal to the sliding direction reduced friction to 63% in boundary and mixed friction, due to decreasing asperity contact. Angular grooves, chevrons, and crosshatch grooves led to smaller friction reduction. Grooves parallel to the sliding direction led to an increase in friction, compared to smooth samples. At the full film regime, surface texturing led to an increase in friction compared to the smooth surface. The stroke was 26.8 mm.

In [71] they found that the textured surface led to an increase in the thickness of the oil film in the mixed lubrication regime to about 20 nm. This small increase caused a reduction in friction force up to 41%. In full-film lubrication, surface texturing led to a decrease in oil film thickness and increase in friction force. Based on [70], the grooves normal to the sliding direction were tested.

The effect of each oil pocket on the change in oil film thickness was experimentally studied [72]. At low speed, the friction force falls abruptly when the pocket leaves the contact zone, and it decays to the untextured case. For higher speed, friction was reduced in a stepwise manner. The effect of all pockets is summarized, and friction reduction can be high. In the mixed lubrication regime, the friction is very sensitive to oil film thickness increase.

In [73] Vlădescu et al. found that, due to wear, contact progressed to mixed and boundary lubrication regimes. Therefore, the positive role of surface texturing increased. The coefficient of friction and cylinder wear both decreased to nearly 70%. They found that both friction and wear were monotonically reduced as oil volumes increased. The only exception was that when the oil pockets were larger than the contact area the coefficient of friction increased due to the collapse of the lubricant film. However, this did not affect the wear changes—Figure 7. The problem is that non-typical materials were used—the fused silica specimen co-acted with a steel pad. Typically, cylinder liners are made from cast iron, and their wear levels are small. The duration of the test was 4 h.

Vlădescu et al. [74] looked at the mechanism of friction reduction due to surface texturing. During the entrainment of the groove-oriented transverse to the sliding direction, cavitation bubbles were observed within each groove, which supports the mechanism of “inlet suction” [75]—texture draws lubricant into contact. Textured surfaces can control oil consumption.

In an experiment performed by Vlădescu et al. [76] the depth, width, and areal density of rectangular oil pockets varied independently to obtain optimum parameters. It was shown that in the boundary lubrication regime, oil pockets should be deep, wide, and densely spaced; because of this, the lubricant volume should be increased. However, there should be a limit on oil volume. In the transition between mixed lubrication and the full friction regime, the dimples should be narrow and sparsely spaced. Oil pockets should be present very close to reversals, but not at a point of reversal. The dimple pattern should vary along the stroke of the piston (Figure 8).

Profito et al. [77] obtained a friction reduction due to the creation of rectangular dimples of 80 µm width, 8 µm depth, and the distance between the dimples was 1100 µm.

Ma et al. [78] studied the effects of macro-scale grooves created on cylinder liner surfaces on friction reduction. The pit area ratios were 10, 25, and 50%. The depths of the dimples were large, near 0.1 mm. The stroke was 100 mm. The best results were obtained for the smallest dimple density; however, the worst results were obtained for the largest ones. The reason for the highest friction reduction for the area density of 10% according to the authors of [78] is the pressure perturbations as a result of micro-hydrodynamics, made by the wedge effect at the inlet of the introduced groove. The pit area ratio of 50% contributed negatively to increasing the hydrodynamic friction force. The effects of texturing were greater at smaller normal loads.

Yousfi et al. [79] compared the tribological performance of the cylinder liner after helical slide honing with those having circles and ellipses. They found that ellipse patterns oriented parallel to the sliding direction led to the highest reduction of the friction force compared to the honed surface.

Johansson et al. [80] created elliptical oil pockets of relatively large diameters, between 1.9 and 2.9 mm, and the pit area ratio was near 30%. The depths of the two types of texture were different: nearly 20 and nearly 90 µm. The dimples were created using CNC. The coaction between the cylinder liner and the oil control ring was tested. The friction tests, under lubricated conditions, were performed using an eccentric reciprocating tester. The stroke was 30 mm. It was found that surface texturing led to a decrease in friction and wear.

Guo et al. [81] studied the effect of cylinder liner texturing on friction and wear in a reciprocating motion. Samples and counter-samples were made of cast iron. The sliding assembly was lubricated with oil without additives. The stroke was 100 mm. Isolated dimples and chevrons were created (Figure 9). The depths of the dimples were sized at 0.2 mm. Unfortunately, the reference sample was not tested. The experiments were carried out with three loads. Generally, chevrons obtained the best wear performance. In addition, at low load, this texture led to the smallest coefficient of friction. In [81], changes in three-dimensional surface topography are characterized by four parameters, for example, rms. height Sq and rms. slope Sdq [82] characterized the wear of the liner samples.

Rao et al. [83] analyzed the tribological performance of the cylinder liner with thread grooves of 1, 2, 3 and 4 mm widths under reciprocal motion. The 2 mm groove structure was found to improve wear performance at smaller, while 3 mm at higher speeds. The grooves had a depth of 0.2 mm. They were made by the photochemical etching process.

In [84] they also studied the tribological impacts of groove inclination; concave pattern was also studied—Figure 10. They found that both types of texturing led to friction reduction. Among surfaces with grooves, the inclination angle of 30° guaranteed the best tribological performance: the coefficient of friction decreased to 58%.

In [85] samples of nodular cast iron cylinder liners were made by encapsulation of WS2 in the dimples on the surface of the liner. The experiments were carried out using a home-made reciprocation tester. The stroke was 90 mm. The depths of liners were comparatively high, larger than 150 μm. The dimple diameters were between 0.4 and 1 mm. Detailed information on the pit area ratios were not given, they increased when dimple diameters increased. Tests were carried out using step loading conditions. The effects of surface texturing and the presence of solid lubricant presence led to an improvement of frictional and anti-seizure properties compared to behaviors of untextured and only textured surfaces.

Miao et al. [27] compared the effects of the piston ring and the cylinder liner texturing on the tribological performance of the piston ring assembly. The surface of the cylindrical liner contained grooves, while the piston ring contained isolated circular dimples. However, the depth of the textured surfaces was high (120 μm). The pit area ratio of the textured piston ring surface was 15%, while that of the liner texture was 5%. Although the improvement in friction and contact resistance was higher for the textured piston ring surface, the effects of accumulating wear debris were stronger for the textured cylinder liner surface. The best results were achieved after combining texturing of both surfaces: cylinder liner and piston ring, the coefficient of friction decreased by 44.5%. Texturing both surfaces was analyzed rarely since it is related to an additional cost. Wos et al. [86] found that both surface textures caused small coefficient of friction values and fluctuations when the pit area ratios of both surfaces were low.

Yin et al. [87] compared experimentally the performance of the surface of the cylinder liner after plateau honing and the cylinder liner textured by laser technique. The laser-textured surface measured including dimples was characterized by smaller values of the Rk and Rpk parameters, but higher values of the Rvk parameter compared to the plateau-honed cylinder liner. Therefore, the surfaces after laser texturing were closer to the ideal surface (Figure 11). Due to LST, the friction within one stroke was reduced. The higher reduction was achieved at a lower temperature (up to 38%) when friction was hydrodynamic due to lubricant rheology. However, at higher temperatures (mixed friction), the maximum reduction was 6%.

Xu et al. [88] created three types of texture on a boron cast iron cylinder liner: grooves, dimples, and grooves+dimples. The normal load was 210 N, stroke was 80 mm. The depths of the dimples and lines were 8 μm, while their diameters were 1 mm. There were the following dimple densities: 6, 12, and 24%. The groove and dimple structures showed better tribological performance than untextured liners. However, the groove+dimple configurations led to high liner wear. Among structures containing dimples, the best antifriction and anti-wear performance was obtained for a density of 12%. The textured surfaces lubricated with bio-oil with MoS2 microsheets presented optimal friction reduction and anti-wear properties.

Stoeterau et al. [89] created using a laser similar texture to that obtained in the honing process. Contrary to honing, grooves produced by the laser had constant dimensions, free of folded metal. Initial tests performed using Optimol tribotester with low viscosity oil proved that friction of laser textured surfaces was similar to those created by honing.

Shen et al. [90] used the reciprocating electrolyte jet with the prefabricated mask technique to fabricate dimples on cast iron cylinder liners. The depth of the dimple was 43 μm, and the pit area ratio was 22%. Surface texturing led to a decrease in friction of 12% at a high load of 20 MPa.

Xu et al. [91] found that the dimple presence had marginal influence on the friction force of the oil control ring-liner sliding pair.

The results of friction tests under lubricated reciprocating motion can be helpful for texturing cylinder liners. Among the textured samples, the chevrons led to the highest increase in hydrodynamic oil film thickness, and grooves were the least effective [69]. Nakano et al. [92] found a decrease and an increase in friction when dimples and grooves were created on cast iron surfaces, respectively. Vilhena et al. [93] found that at flat contact, the presence of dimples on a flat surface caused a reduction in the coefficient of friction for a lower sliding speed. However, the effect of surface texturing on the coefficient of friction was small. For higher speeds, surface texturing led to an increase in the coefficient of friction compared to untextured surfaces, especially with a high pit area ratio (27%). The authors explained this behavior by the positive effect of surface texturing in boundary lubrication (lower speed) and disturbances in mixed and full lubrications (higher speed). In [94] Podgornik et al. found that grooves in general led to increased friction in starved lubrication, being an obstacle to the sliding motion. Saeidi et al. [95] found that the dimple diameter and the pit area ratio of the grey cast iron surfaces had the greatest effect on the coefficient of friction. From among various dimple and oval patterns, the oval positioned perpendicularly to the sliding direction (length 0.5 mm. width 0.1 mm, depth 50 μm, density 5%) gave the smallest coefficient of friction. The beneficial effects of surface texturing were observed near full-film lubrication conditions. Fang et al. [96] found that crossing line patterns achieved a smaller and more stable coefficient of friction compared to the array of dimples. Lu et al. [97] studied the effect of surface texturing in line contact during reciprocating lubricating sliding. The presence of square dimples led to a 15% reduction in friction in boundary lubrication. They also revealed that the converging triangular [98] and square [99] dimples led to a reduction in friction. Wos et al. [55] found that sandglass-shaped oil pockets behaved well in reciprocation lubricated motion, especially with the pit area ratio of 5%. They also revealed [100] that the effect of the density of dimples on the reciprocating motion was more visible at a higher temperature of 80 °C, compared to a temperature of 30 °C.

Guo et al. [101] tested the influence of cylinder liner surface texturing on wear using an engine simulator. It was found that the highest wear resistance was obtained for the surface with isolated dimples. In this work, the effect of dimples, grooves, or their combination was studied. The depths of the features were between 0.2 and 0.3 mm, which seemed to be high.

Peng and Huang [102] powered the real engine with an electric motor to ensure conditions similar to those of the real engine. Cylinder liner B had grooves of comparatively high depth (0.3 mm, width 3 mm), cylinder liner C had isolated dimples of similar depth and diameter of 1 mm, while cylinder liner D had a combination of grooves and dimples (Figure 12). The presence of isolated dimples (cylinder C) minimized the risk of scuffing. However, serious scuffing occurred for the untextured liner (after honing) and for the liners with grooves. Cylinder C was also found to have the highest wear resistance. The duration of the test was 5 days.

Rao et al. [103] analyzed the tribological behavior of cylinder liners with thread grooves of 1, 2, 3, and 4 mm width and 0.2–0.3 mm depth, in a real engine powered by an electric motor. The inclination angle was 30°. They found that the 3 mm structure ensured the best tribological properties: the friction coefficient decreased by 31%, the contact resistance increased by 33%, and the sealing performance was improved by 14% compared to the untextured liner. The electrical conductivity discrepancy of the lubricating oil relative to the metal was the operating principle of the contact resistance measurement.

Yin et al. [104] analyzed the effects of various arrays on the motor torque of the gasoline engine. The dimple radius was 35 μm, the depth was 8 μm, and the pit area ratio was 10%. Figure 13 shows arrays of dimples. The motor torque was the smallest for the square array, then stretching along the liner axis array and the stagger array. The maximum increase from the square array to the stagger array was 3.9%.

Yin et al. [43] obtained smaller friction torque of an internal combustion gasoline engine with a dimpled cylinder compared to engines with plateau-honed cylinder surfaces.

Typically, dimples are located near the TDC of piston rings. However, Urabe et al. [105] created oil pockets at the midpoint of the cylinder liner (Figure 14). On the test rig, they used the crank angle mechanism of the real engine. A friction reduction of 30% was obtained. In an area subjected to surface texturing, oil film thickness decreased. The depth of the dimples was small, nearly 2 μm, the diameter of 0.5 mm was less than the width of the first ring, and the pit area ratio was large, nearly 50%. The effect of dimple shape (circular, hexagonal, or square) was analyzed. The idea of texturing was to reduce the area of contact, leading to a reduction in the oil film thickness and friction. In the dimpled area, the thickness of the oil film was diminished.

### 2.2. Tests Using a Fired Internal Combustion Engine

These types of tests are more expensive than laboratory simulations. However, the cylinder liner co-acts with a few piston rings and the piston skirt (Figure 14). It is difficult to obtain good repeatability of results because there are many elements in internal combustion engines. The results are not as substantial as in laboratory tests. Piston ring texturing caused a reduction in the friction force of about 50% and fuel consumption decreased only by 4% [21,22]. In these types of tests, there are more output variables than in laboratory simulations: fuel consumption, oil consumption, exhaust emissions, torque, effective power, etc.

Ergen et al. [106] studied the effects of surface texturing on the oil consumption of diesel engines. In variant 1, only the zone near the top dead center was subjected to texturing. The oil pockets were oval in shape (length 3 mm, depth 10–20 µm, width 50–80 µm). In variant 2, the entire liner surface was subjected to texturing, and only for of the top reversal distances between the oil pockets were (the pit area ratio was smaller). Variant 2 led to higher oil consumption than variant 1. Both laser textured variants and especially variant 1 assured smaller oil consumption than surface after plateau honing, of higher oil capacity, which was estimated by the present authors based on parameters Rvk and Mr2. Figure 15 shows the configurations of the oil pockets.

High oil consumption is undesirable, as it causes high emissions from the engine. Oil consumption is related to the volume of the lubricant reservoir in surface topography, called oil capacity or oil retention volume [48,107,108,109]. Typically, oil consumption is higher for higher oil capacity. The oil capacity is higher for a higher roughness height and for the same surface amplitude is higher for plateaued surface compared to a surface of Gaussian ordinate distribution [48,108]. However, the surface cannot be very smooth, due to the tendency to scuff [110,111,112]. There is a problem with the correct estimation of the oil capacity. The application of the parameters of the Sk family can lead to serious errors. The best method is to normalize the material ratio curve, its rotation of 45° angle, and find the highest point. The oil capacity is the volume located below the transition point (hatched area in Figure 16). Other proposals for oil capacity calculations are given in [113,114].

Hua et al. [115] achieved reductions in oil consumption and emission of the internal combustion engine by about 50% due to additional laser surface texturing compared to plateau-honed liner surfaces, however, power, torque, and fuel consumption were stable.

Hua et al. [116] tested three arrays of dimples on the cylinder liner surface (Figure 17). The behaviors of laser textures liner surfaces were compared with those with plateau-honed cylinder surfaces. Surface LST-1 and LST-3 caused a smaller oil consumption compared to the performance of the standard honed surface, contrary to surface LST-2, leading to the highest oil consumption due to dimple presence on the whole surface. The kinds of textures led to a decrease in oil consumption of up to 45%. The highest decrease in fuel consumption (up to 5%) was achieved for surface LST-3, which was recommended.

Similar results were reported in [117]. However, in [116,117] the oil capacities of textured and untextured surfaces were not compared. Information that the surface before texturing was subjected to additional honing to reduce the height of the roughness is substantial [117].

Brinkman and Bodschwinna [118] tested engines of Volkswagen vehicle equipped with plateau-honed cylinder liner surfaces and surfaces with oil pockets on the engine test bench. The second surface was characterized by a lower bearing roughness height with dimples being oil reservoirs. The surface with separated oil pockets led to less oil consumption and smaller volumetric wear during running-in than plateau-honed texture. Volumetric wear shown in Figure 18 was calculated based on a comparison of material ratio curves before and after tests, therefore, wear level was related to the measurement area.

Golloch and Merker [119] compared the functional behavior of a four-stroke Diesel engine with honed and laser structured cylinder liner surfaces. The oil pockets had a length of 3 mm, a width of 40–60 µm and a depth of 10–25 µm. The laser-patterned surface showed a lower friction mean effective pressure (FMEP), smaller wear and an oil consumption and increase in the oil film thickness than the honed surface.

Rahnejat et al. [120] created an oil pocket pattern on aluminum cylinder liner, coated with Nikasil from a gasoline four-stroke engine (Figure 19). Surface texturing led to an increase in torque up to 4.5% which was probably related to increasing oil film thickness in mixed lubrication regime.

Yin [121] found that laser texturing led to a decrease in the fuel consumption of the internal combustion engine at low speeds of up to 4.62% and to a reduction in oil consumption of up to 34%.

Howell-Smith et al. [122] believe that texturing of the piston ring leads to the deterioration of its sealing property. On the contrary, the possibility of boundary lubrication cylinder liner texturing near the TDC of the piston ring would improve operating parameters. They used laser-etched cylinder liners and indented oil pockets (Figure 20). Both types of texturing led to a power increase of up to 4%; this effect increased with increasing speed for indented dimples, contrary to oil pockets made by laser etching. According to the authors, the laser-textured pattern encouraged the flow of lubricant into the region of high contact pressures during the reversal. When the rotational speed increased, the behavior was similar to that of an untextured surface. However, the indented imprint pattern acted more like a reservoir of lubricant.

It was found from tests of diesel engines that surface texturing of the cylinder liner at the mid-stroke caused a decrease in fuel consumption of heavy-duty vehicles from 0.6 to 3.2% [105]. The friction reduction on the test rig was 30%. Surface texturing did not cause increased oil consumption, and decreased engine durability and cylinder and ring wear. However, increasing the depth of the dimple could lead to higher oil consumption.

Cylinders with Nikasil coating were subjected to texturing by burnishing technique [123]. The dimples had diameters of 0.25–0.35 mm and depths of 4–6 µm—Figure 21. Additionally, in one case, surface texturing was combined with a DLC coating. Surface texturing resulted in an increase in power, and torque improvement mainly for high engine rotational speeds. The best results (power increase of 5.8%) were achieved for the combination of surface texturing and DLC coating.

Rao et al. [29] studied the effect of the texture of the liner surface texturing on wear, Nox emission, and engine vibration. Cylinder liners were made of boron cast iron. The thread grooves on the CNC machined liner surface had widths of 1, 2, and 3 mm (Figure 22) and depths of around 0.2 mm. The alternative was to combine these grooves with dimples made on the first piston ring by laser. The diameter of the dimple was 1 mm, and the depth was 0.115 mm. It was found that the presence of the grooves could reduce the wear of the liner and the ring, the Nox emissions (more than 10%), and the violent vibration of the engine. The best parameters were obtained for grooves of 2 mm width. The test duration was 6 h. The storage of lubricant within valleys caused small wear on the analyzed system. Smaller Nox emissions can be explained by better sealing of the textured assembly, as the burned gas was expelled more quickly. The stored lubricant had a positive effect on damping.

Laser honing allows for the formation of oil channels of desired dimensions leading to improvement of lubrication. Zhang et al. [124,125] found that laser honing caused a decrease in oil consumption to more than half, wear of piston ring and cylinder liner [124] and exhaust emission [125] to about half, compared to plateau honing.

### 2.3. Computer Simulations

Simulation of loads, temperatures, and movement within the combustion chamber is very difficult but possible using novel computers and software. One can also model the surface topography of textured surfaces. Computer modeling should be the first stage in research, and the laboratory or better engine tests should be the second stage. The application of modeling is much cheaper than experimental research. Only the best results should be subjected to experimental validation. Because mixed friction between cylinder liner and piston rings takes place at reversal points, the lubrication and surface contact are typically analyzed. Often the average flow model developed by Patir and Cheng [126] and the elastic contact developed by Greenwood and Tripp [127] are applied. In some works, only hydrodynamic lubrication is simulated; they can be used in mid-stroke modeling.

Organisciak et al. [128] analyzed the starved hydrodynamic lubrication between the piston ring and the cylinder liner. They found that grooves led to an increase in the oil film thickness and more lubricant passed in the grooves.

Takata et al. [129] analyzed the co-action between the cylinder liner and the piston ring under mixed friction. They analyzed oil flow and contact between asperities and found that deeper features and more transverse grooves, as well as a higher pit area ratio of dimpled and grooved surfaces, leading to the greatest friction reduction.

Rahnejat et al. [120] found that additional oil pockets helped to increase the minimum oil film thickness.

Caciu et al. [130] simulated hydrodynamic contact between the cylinder and the piston ring, represented by flat surfaces. Rhomboidal cavities with trapezoidal profiles led to the greatest reduction in the coefficient of fiction.

Yin [131] found that although surface texturing plays the main role in the reduction of asperity contact near the TDC (top dead center) and BDC (bottom dead center), it still has a positive effect on hydrodynamic lubrication in other areas.

This researcher [121] predicted an increase in minimum oil film thickness in the middle part of the stroke and a decrease in the friction peak of up to 30%.

Yin et al. [132] simulated lubrication between the cylinder liner and the piston ring, analyzing the Reynold equation revised by Patir and Chang and contact between asperities using the Greenwood–Williamson model. Due to texturing, mixed lubrication existed only near the reversal locations of the piston ring. They obtained the optimum dimple parameters: radius r_p_ of 30–60 µm, pit area ratio of 0.2–0.4, and depth h_p_ of 3–6 µm. In [133] they found that the tribological performance of the textured surface of the cylinder liner depends on the combustion mode. They also considered the effect of isolated dimples on the functional properties of the piston ring pack [134]. They analyzed hydrodynamic lubrication and contact between asperities. They found that due to surface texturing (Figure 23) the maximum asperity pressure, the average friction force and friction losses can be reduced.

In [104] they studied the tribological effects of the dimple array (Figure 24). Lubrication was the best (minimum film thickness was the highest) for the square array and the worst for the stagger array. The results were experimentally validated.

In a simulation of mixed lubrication between the cylinder liner and the piston ring, Yin et al. [43] found that the liner surface with additional dimples caused an increase in oil film thickness and a decrease in asperity contact, leading to reduced friction, compared to plateau honed surface.

Zhou et al. [135] analyzed the lubrication using the Reynolds equation between the textured cylinder and the piston ring at various speeds. The theoretical model of oil film thickness and load-carrying capacity was developed. They found that when speed increases, the optimum area density decreases. Texturing with variable parameters produced a thicker film than invariable texturing (Figure 25). The highest oil film thickness was obtained for variant a, followed by d, c, and b.

Tomanik [136] found that the surface texturing of cylinder liner has the potential to reduce friction and wear by generating hydrodynamic support when the liner co-acts with a flat, thin oil control piston ring, however, the liner texturing effect of the liner texture is marginal during contact with the barrel top ring.

Based on lubrication theory, Zhan and Yang [64] obtained the following parameters for the textured cylinder liner: the ratio of the depth of the dimple to diameter 0.1–0.2, pit area ratio 20–40% and θ angle (Figure 2) 45°.

Checo et al. [137] developed a model based on the Elrod–Adams [138] model of lubrication and cavitation. They found that surface texturing of the cylinder liner was detrimental in non-conformal contact with the piston ring. For quasi-conformal contact, a friction reduction of up to 73% due to surface texturing was predicted. The dimples had an ellipsoidal shape (Figure 26).

The superiority of multi-scale grooves density of 10–25% over 50% for reducing friction was found in [139,140]. The proposed density associated with a depth-to-width ratio of 0.06–0.14 yielded the best results [140]. Recently, Mohamad and Kamel [141] studied the effects of parabolic macro-scale grooves (Figure 27) on cylinder liners on friction reduction. Similarly to [139,140], they analyzed hydrodynamic lubrication and asperity contact pressure. On the basis of analysis, they selected the optimum dimensions of the grooves. The presence of grooves caused decreases in hydrodynamic friction and asperity contact friction. The teaching–learning-based optimization algorithm was used to obtain the dimensions of the grooves. Surface texturing would be more beneficial at TDC than at BDC. The results obtained are: groove depth h_p_ 0.01 mm, groove width 2.46 mm, groove area density 49%, and groove aspect ratio 0.002.

Ma et al. [78] modeled the contact between the cylinder liners with micro-grooves and the piston ring. They studied hydrodynamic lubrication and contact between asperities. They analyzed the following dimple densities: 10, 25, and 50% with depth near 0.1 mm. The pit area ratio of 10% led to the greatest reduction in friction force. Parabolic and triangular grooves behaved better than rectangular ones. This model was validated by experiment.

Zavos and Nikolakopoulos [142] analyzed full film lubrication between the cylinder and piston ring, and the contact of asperities was neglected. Wave-cut and honed cylinder liners were modeled (Figure 28). Both types of cylinder texturing were found to cause a friction reduction of up to 22% and an increase in oil film thickness to 2.1%. However, the positive effect of honing was greater. Further friction reduction can be obtained by texturing the piston rings.

Biboulet and Lubrecht [143] modeled the starved contact between the piston ring and the cylinder. They found through simulation that the impact of liner surface texturing depends on the type of rings. Deep and wide valleys have a detrimental effect on the load-carrying capacity when the ring has a small radius of curvature, whereas dimples on the liner surface can lead to an increase in the load capacity when the piston ring has a large radius of curvature. They found that the location of the grooves and their dimensions had a greater impact than the shape of the groove (triangular or sinusoidal).

Gu et al. [144] analyzed the sliding contact between the cylinder liner and the piston ring under starved lubrication, considering the oil supply. They studied hydrodynamic lubrication and contact of asperities. Surface texturing of the cylinder liner can have steady beneficial effects, when the minimum oil film thickness or external load were fixed, on tribological properties. However, the positive effects of the textured ring are unstable in starved lubrication.

Morris et al. [68,145] analyzed through numerical simulation the effect of various chevron patterns on the cylinder liner surface on the friction decrease. Although the reduction in test rigs due to surface texturing was higher than 10% [68] this reduction in the fired engine was marginal, nearly 1%. The models developed in [68,145] were experimentally validated.

Profito et al. [146] simulated lubrication between the piston ring and the textured liner by analyzing hydrodynamic lubrication and contact between asperities. The model shows an increase in friction under mixed lubrication as the dimple entered the contact followed by a decrease in friction as the pocket left the contact to gradually decay to the steady value. The results of the simulations were experimentally confirmed under pin-on-disc configuration.

Profito et al. [77] developed a model of mixed lubrication between the piston ring and the textured cylinder liner. This model was experimentally validated. It was found that, due to liner surface texturing, the friction decreased due to the reduction in asperity contact.

Garcia et al. [147] in modeling analyzed also hydrodynamic lubrication and contact between asperities. They found that surface texturing of the liner led to a reduction in the asperity contact force of 20%, a total friction force of 5%, and an increase in minimum oil film thickness of 4%, so the power loss can be reduced. The numerical model was experimentally validated. The depth of the groove was 1.2 µm, the radius of the groove was 50 µm and the distance between the grooves was 150 µm.

Jang simulated co-action between the piston ring and cylinder liner with dimples on the honed surface of length 3 mm, width 60 µm, and depth 20 µm near the TDC of the first piston ring [148]. The piston ring profile was considered to be a parabolic curve. Hydrodynamic lubrication and contact between asperities were considered. The minimum oil film was thicker for a laser-patterned surface than for a honed surface. The thicker film was associated with less wear and friction.

The results of the modeling of surface contact in reciprocating sliding can be helpful in analyzing contact between the piston ring and the cylinder wall. Shen and Khonsari [149] based on the modeling of recommended trapezoidal shapes in bidirectional sliding.

## 3. Discussion

The role of surface topography of cylinder liners increased recently, because the growth of loads, speeds and temperatures caused the decrease in the oil film thickness [150]. However, the limitation of emissions complicates the situation, because there is a need to decrease oil consumption related to the oil capacity [113,114]. There are also demands for a decrease in fuel consumption, which is directly related to a reduction in friction losses. These losses can be reduced by increasing the amount of oil available in the piston ring pack, which is related to the increase in oil consumption. The creation of oil pockets on the cylinder wall can give a better compromise between friction losses and oil consumption, leading to a decrease in friction losses without increasing oil consumption. This approach can lead to the use of oil additives, so the reliability of the engine can be maintained. Additional texturing of the cylinder liner can be used to obtain various shapes and dimensions of oil pockets, which is not possible to achieve using plateau honing. However, oil consumption is not important in high-performance or marine engines.

The best procedure is at first to model the tribological impacts of cylinder liner texturing and experimentally test only the optimum solutions. However, this procedure was rarely applied, only the authors of considered papers: [43,64,68,77,78,121] combined modeling and experimental investigations.

In modeling, the following output parameters were typically used: oil film thickness and coefficient of friction; load carrying capacity and asperity contact pressure were also applied. The presence of dimples typically led to an increase in oil film thickness and a reduction in asperity contact force near piston ring reversals (TDC and BDC). These changes caused a reduction in friction. The authors of the works [64,121,131,135,137,142] found that the oil pockets had also a positive effect on the hydrodynamic lubrication, which existed in the midpoints of the stroke. In the models, oil consumption was not predicted. Most of the models used were simple, for example, they are based on the old Greenwood–Tripp [127] model.

Typically, the tribological impacts of spherical dimples and grooves were modeled. The dimples had a depth of up to 10 µm, a pit area ratio of 10–40% and a ratio of dimple depth to diameter of 0.1–0.2. Unfortunately, the oil capacity was not calculated. Only in works [104,135] the array of dimples was taken into consideration. Oil pockets of other shapes were analyzed rarely [68,130,137,145,149], although various forms of textured surfaces can be simulated.

The highest number of papers was found for experimental research using test rigs. However, it is very difficult to simulate a co-action between the piston ring and the cylinder liner. First, the test should be completed in a lubricated reciprocating motion. Second, the detail of the cylinder liner should co-act with the details of the piston ring. In experimental investigations using reciprocating test rigs typically the effect of oil pockets’ presence on the coefficient of friction was studied. In some cases, the wear levels of co-acting pairs were considered. The advantage of this kind of research is the possibility to analyze the friction coefficient course within one stroke; because of it, the impact of each dimple presence on frictional resistance can be studied. However, this possibility was rarely studied [55,72,76,100]. The correct simulation of phenomena occurring in a fired engine is a serious problem. Mechanical, thermal, chemical, lubrication, lubricant, and materials factors should be considered, as well as third bodies. For example, the chemical nature of tribofilms formed in a fired engine and in a reciprocating tester is different [151]. There is a problem with the available stroke length. Because it is difficult to change the load over the course of the stroke, only one stroke position can be replicated [152]. Of course, testing using test rigs can reduce the cost of research; however, they can give erroneous results when applying results to real engines. The researchers tried to simulate test conditions from the real engine. However, elevated temperature tribological tests were seldom conducted [66,87].

Similarly to computer simulations, typically isolated dimples or grooves were created on cylinder wall surfaces. Dimples were characterized by dimensions and pit area ratio, similar to grooves, which can sometimes be characterized by the distance between them, rather than their density. The oil capacity typically was not used for the descriptions of textured surfaces. However, Vlădescu et al. [76] found that friction and wear were reduced when oil volumes increased.

There were two types of grooves/dimples. In the first case, they had small depths, typically up to 10 µm. The pit area ratio was small, between 5 and 15%, and for circular dimples, the ratio of the depth of the dimple to diameter was 0.1 or smaller [62,67,70,77,88]. In the second case, the depth of the dimple was larger, 0.1–0.3 mm [78,81,83,84]. Macro-scale texturing, with an oil pocket depth of 0.1 mm or higher, was dedicated to cylinder liners of marine engines. Surface texturing of the cylinder liners led to a decrease in the friction force. According to [70,95], grooves (ovals) should be located perpendicularly to the sliding direction. However, a certain inclination of the grooves was recommended in [84]. The shifts between rows of dimples (Figure 2) had a positive effect on reducing friction and wear [63,64,65].

In most cases, surface texturing led to a decrease in the coefficient of friction and wear. However, Tomanik [60] found a marginal impact of oil pockets. The highest reductions in friction and cylinder volumetric wear were near 70% [73].

The effect of varying the pattern of oil pockets along the stroke was studied rarely. Vlădescu [76] found that near reversals dimples should be deep, wide, and densely spaced, but in the midpoints of the stroke narrow and sparsely spaced.

The effects of tests obtained using a real engine powered by an electric motor are better and similar to those of a fired engine than those obtained from reciprocating sliding. In [101,102,103], it was found that cavities of a large depth (0.2–0.3 mm) on the liner surface led to less wear, a tendency to scuff, and a coefficient of friction than on the untextured surface. Yin et al. [104] recommended a square dimple array with a tribological behavior was better than the stagger array when rows of dimples were shifted. Zhan et al. [63,64,65] obtained contrary results. Urabe et al. [105] created dimples of small depth at the midpoint of the stroke. This pattern led to a reduction in friction at the mid-stroke. These results are interesting since in real engines the dimples are located near the TDC of the first piston ring.

The best information can be obtained from the results obtained from the fired engines. The results presented focused on the surface texturing impact on oil consumption [106,116,117]. One can see that a compromise should be obtained between a decrease in fuel consumption (which should be maximized) and an increase in oil consumption (which should be minimized). Oil consumption is related to oil retention volume (oil capacity), which was only estimated by the present authors. Therefore, oil capacity should be an important parameter characterizing textured surfaces. Due to decreasing the roughness height in areas free of dimples (after honing) oil consumption can be reduced with improvement in lubrication, which can lead to a reduction in fuel consumption. The isolated oil pockets should be located near the TDC of the first piston ring and in the piston skirt (Figure 17), while the pit area ratio in the highest portion of the cylinder wall should be higher than in other places. This trend is in accordance with the results of Vlădescu [76] and Zhou [135]. Additional texturing of the whole cylinder surface led to high oil consumption. However, Urabe et al. [105] obtained a decrease in fuel consumption of up to 3.2% by adding densely spaced dimples (pit area ratio of 50%) of small depth (near 2 µm) only at the midpoint of the stroke. This addition did not cause an increase in oil consumption. However, increasing the depth of the dimple (along with increasing the oil capacity) led to an increase in oil consumption and blow-by.

Yin [121] also reduced fuel and oil consumption, Hua [115] reduced oil consumption, and the authors of papers [118,119] reduced oil consumption and wear due to additional surface texturing. The creation of a dimple can lead to an improvement of the operating parameters, such as torque [120] and power [122,123].

The presence of grooves of 0.2 mm depth caused a reduction in the wear of the liner and ring, emissions of Nox, and violent vibration of the diesel engine [29].

Similarly to creating dimples on the cylinder surface, laser honing caused a decrease in oil consumption [124,125].

However, additional surface texturing of the cylinder liner is seldom performed in the motorized industry, because of the increasing cost. Combining the texturing of the piston ring and the cylinder liner is another possibility [27], however, the cost of machining would be seriously increased. Most likely only high-performance engines have the potential be designed with oil pockets on the cylinder surface. There is less chance to texture piston rings.

Table 1 summarizes research presented in References directly related to surface texturing of cylinder liners.

## 4. Conclusions

The surface texturing of the cylinder liners improves lubrication between the cylinder liner and the piston ring, especially by increasing the thickness of the oil film near reversal points. Consequently, friction and wear of the co-acting pairs can be reduced. A decrease in the resistance to motion is obviously related to an improvement in engine characteristics, such as a decrease in fuel consumption, and an increase in power and/or torque.Correct creation of dimples or grooves on the cylinder surface may lead to a considerable decrease in blow-by and oil consumption compared to plateau-honed cylinder surfaces. This behavior is substantial because smaller oil consumption is related to smaller exhaust emissions. The cylinder surface in the area free of oil pockets should be smooth. Even surface-causing reduced oil consumption can lead to a reduction in fuel consumption. The cylinder liner surface should be additionally textured near the TDC of the piston rings. Texturing the whole cylinder surface or/and too high a depth of oil pockets could cause an increase in oil consumption. An alternative is sparsely texturing the cylinder surface at the midpoint of the stroke. Laser honing also led to a decrease in oil consumption.Circular oil pockets or groves are preferred. Pit area ratio of circular dimples should be between 5 and 15%, the depth of the dimple should be smaller than 10 µm and the ratio of depth to diameter should be less than 0.1. Grooves can be located perpendicularly to the direction of piston ring motion, and of similar depth to dimples, are recommended.An alternative solution is the creation of a dimple pattern of small depth (about 2 µm) and large dimple density at the midpoint of the stroke. This caused a reduction in friction, leading to a reduction in fuel consumption. The other possibility is the creation of inclined macro-grooves with nearly 0.2 mm on the cylinder surface of the diesel marine engine. This solution caused reductions in friction, and wear, better sealing performance, smaller emissions of Nox, and violent vibration of the engine.In future work, the oil capacity should be included in the description of the textured surface. It is related to oil consumption and resistance to motion. The modeling of co-action between piston rings and cylinder walls should consider the deterministic model of contact and co-action between piston rings and cylinder liner in real engines. Tests using rigs in reciprocation motion should correct simulation conditions of co-action between cylinder liners in piston rings in real engines. More tests should be performed using fired engines using an engine test bench, and the effect of liner texturing on engine performance (power, emission, oil consumption, torque, and fuel consumption) should be analyzed.

## Figures and Tables

**Figure 1 materials-15-08629-f001:**
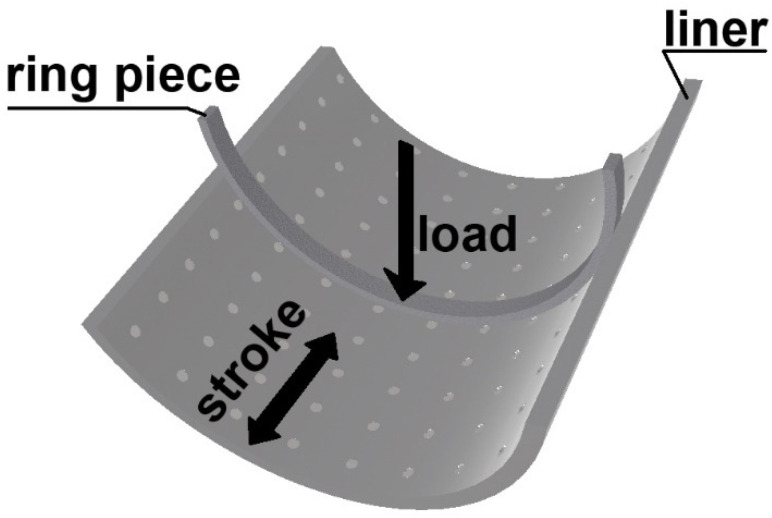
Typical configuration of laboratory tests.

**Figure 2 materials-15-08629-f002:**
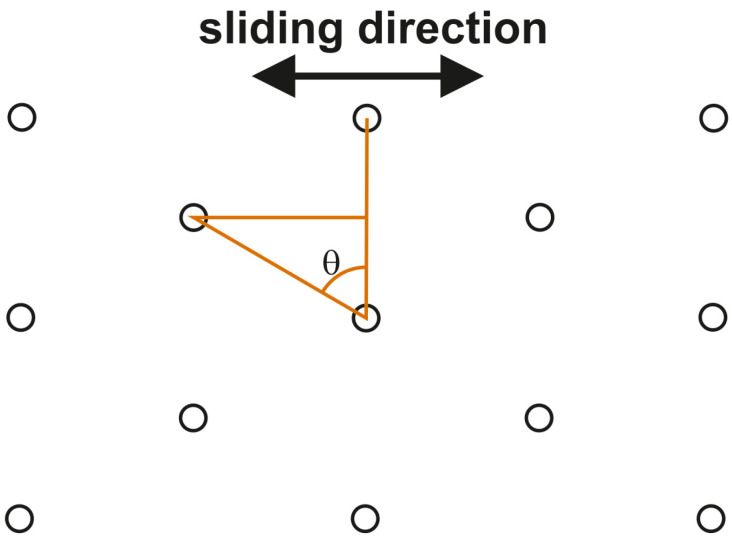
The scheme of laser-textured cylinder liner, after [63].

**Figure 3 materials-15-08629-f003:**
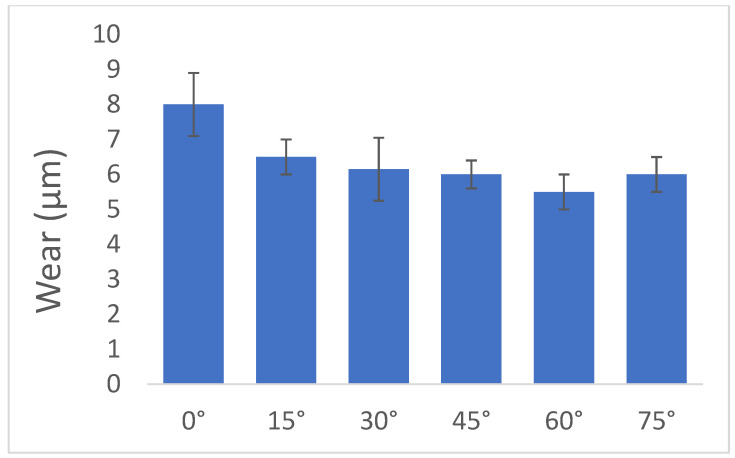
Wear scar depth for the following conditions: test duration 10 h, frequency 4 Hz, stroke 10 mm, maximum contact pressure 85 MPa, after [63].

**Figure 4 materials-15-08629-f004:**
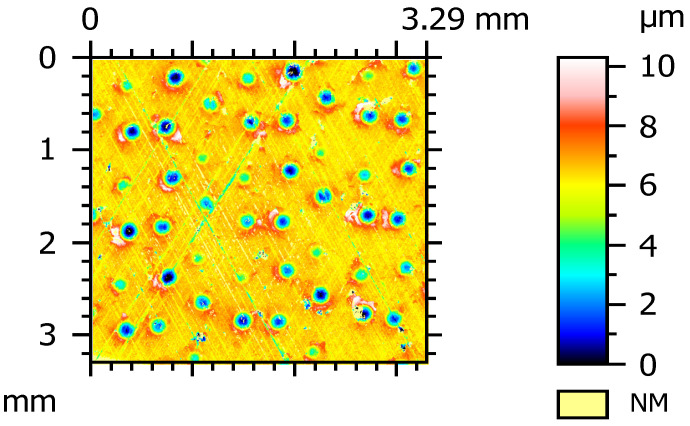
Cylinder liner surface with additional oil pockets tested in [67].

**Figure 5 materials-15-08629-f005:**
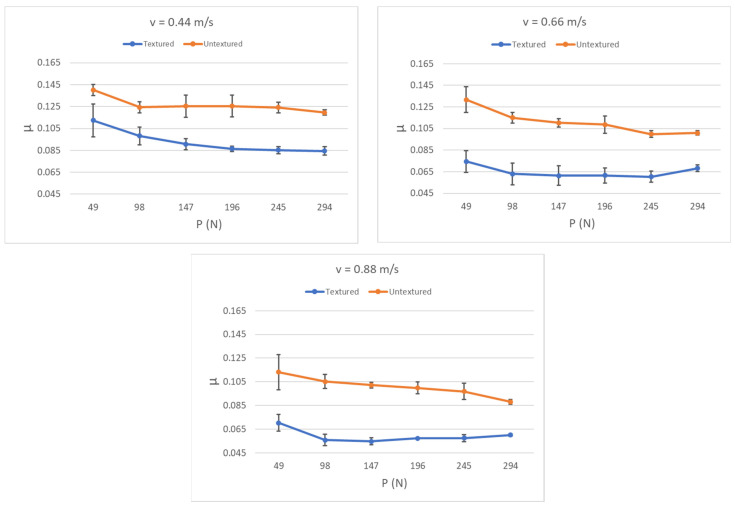
The effect of surface texturing of cylinder liner on the coefficient of friction of sliding assembly under full lubrication, after [67].

**Figure 6 materials-15-08629-f006:**
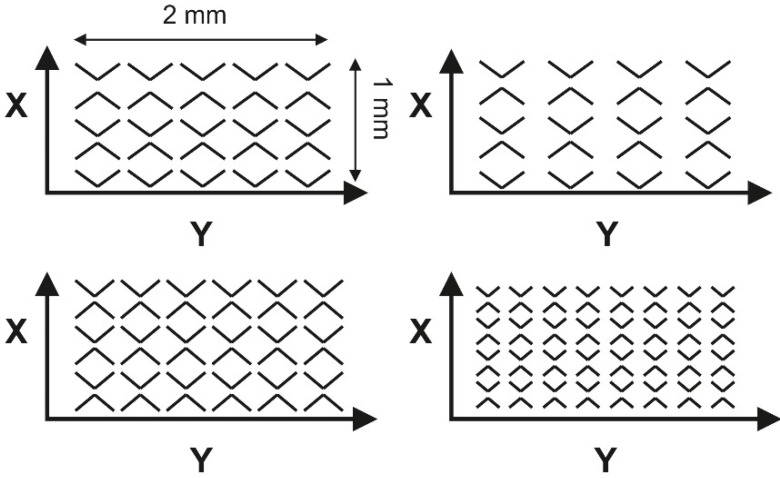
Patterns of chevrons analyzed in [68], X–direction according to cylinder axis.

**Figure 7 materials-15-08629-f007:**
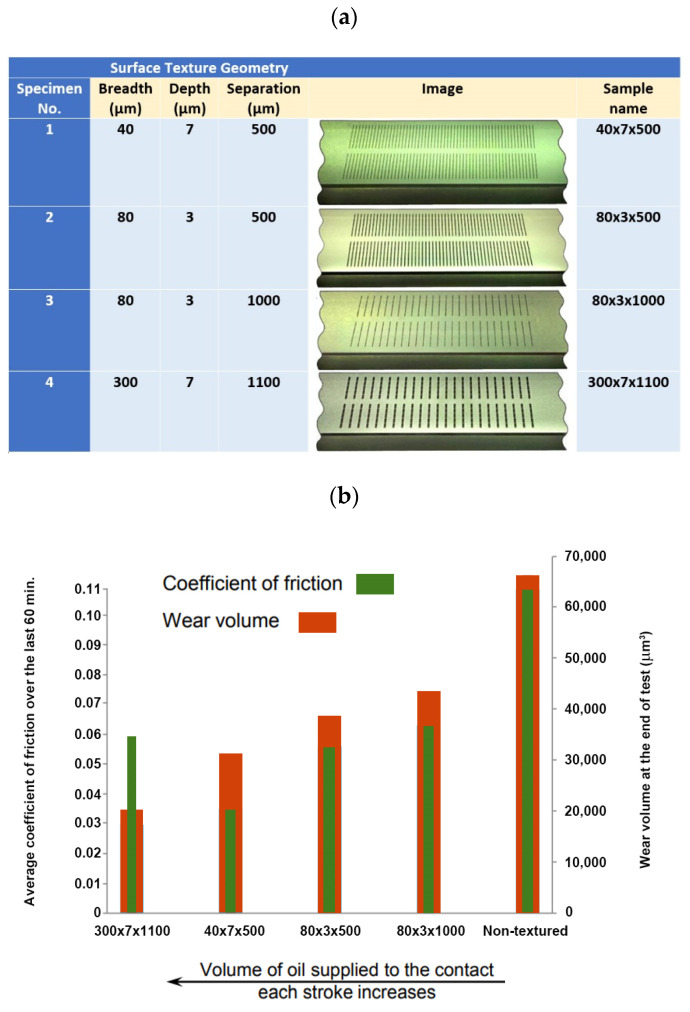
Geometry of texture patterns (**a**), coefficient of friction calculated over the last hour, and final wear for different specimens (**b**), after [73].

**Figure 8 materials-15-08629-f008:**
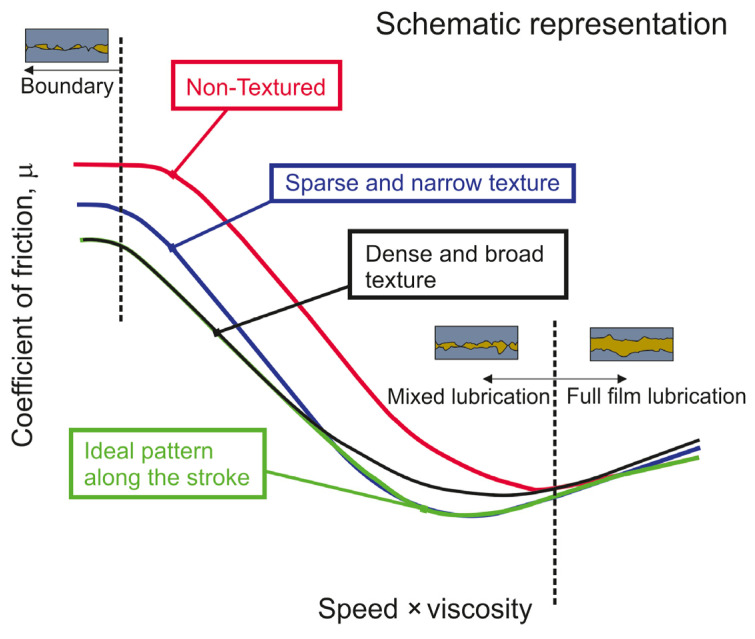
Scheme of criteria for varying the pocket pattern along the stroke, after [76].

**Figure 9 materials-15-08629-f009:**
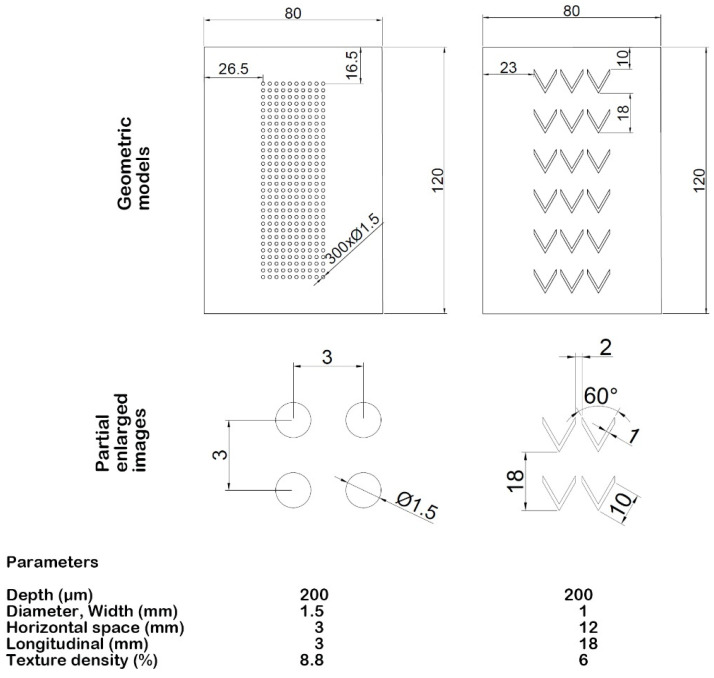
Textured surfaces used in [81].

**Figure 10 materials-15-08629-f010:**
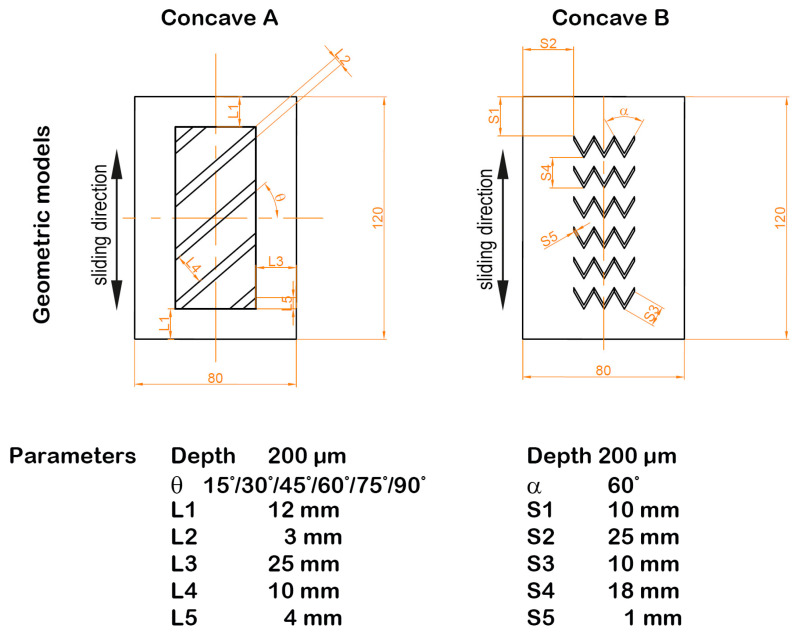
Cylinder samples tested in [84].

**Figure 11 materials-15-08629-f011:**
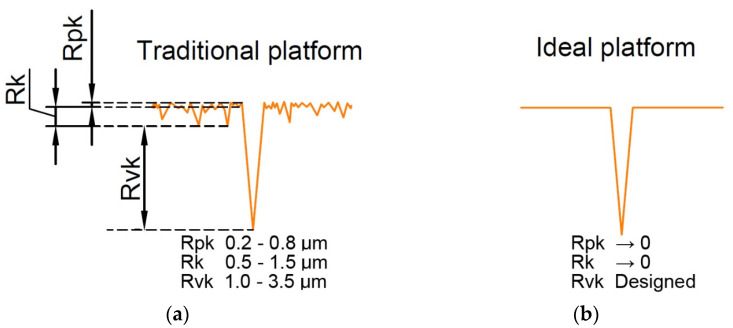
Profiles of the traditional oil pocket (**a**) and the ideal oil pocket (**b**), after [87].

**Figure 12 materials-15-08629-f012:**
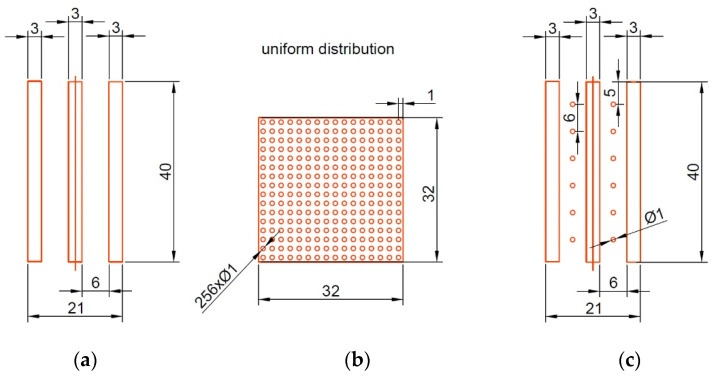
Textured liner surfaces: B (**a**), C (**b**) and D (**c**), after [102].

**Figure 13 materials-15-08629-f013:**
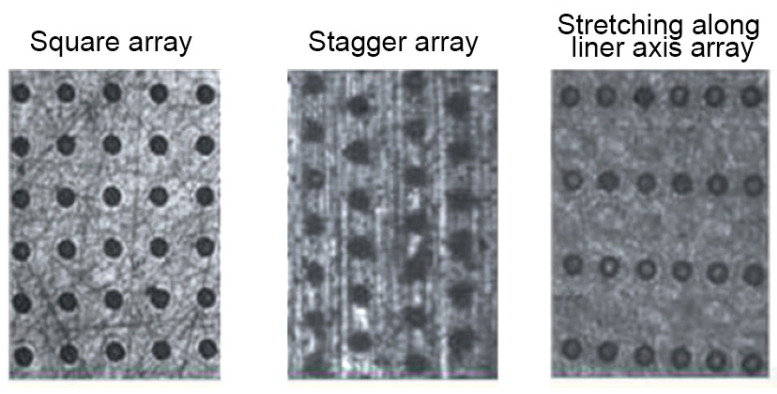
Dimple arrays used in [104].

**Figure 14 materials-15-08629-f014:**
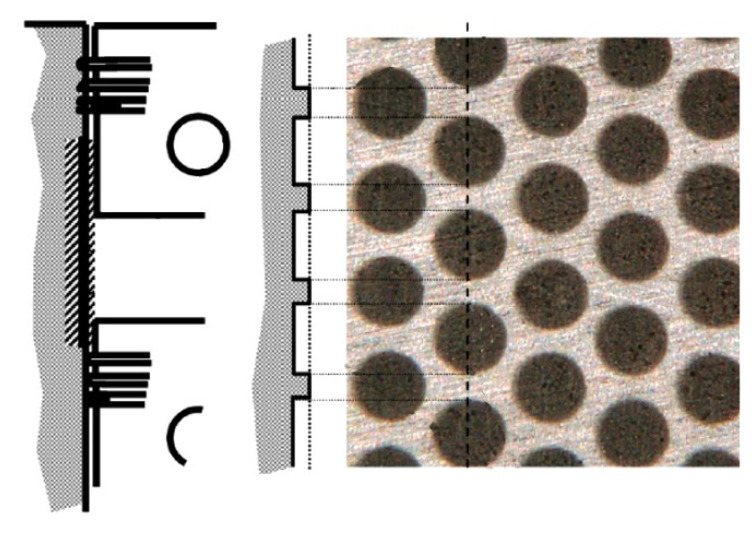
Scheme of surface texturing, after [105].

**Figure 15 materials-15-08629-f015:**
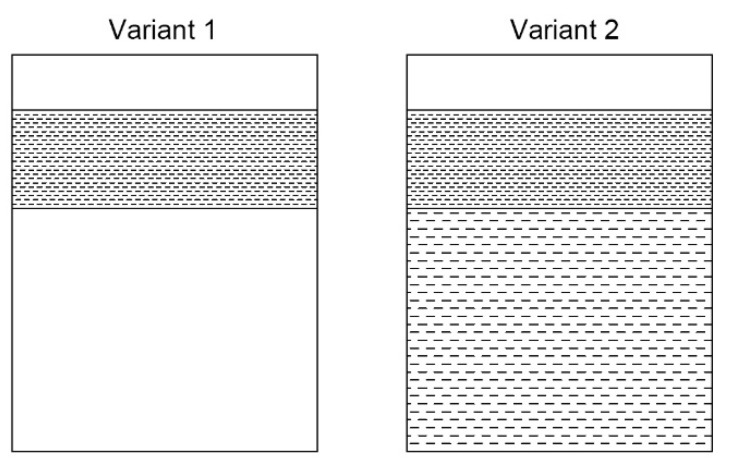
Schemes of cylinder liner surfaces after laser texturing, after [106].

**Figure 16 materials-15-08629-f016:**
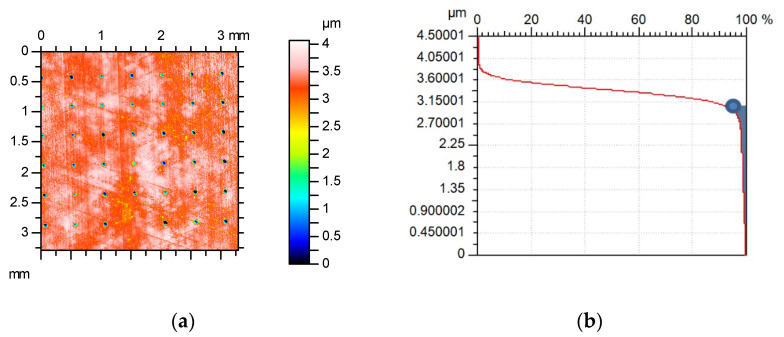
Contour plot of the textured cylinder surface (**a**), its material ratio curve with transition point, and oil capacity (**b**).

**Figure 17 materials-15-08629-f017:**
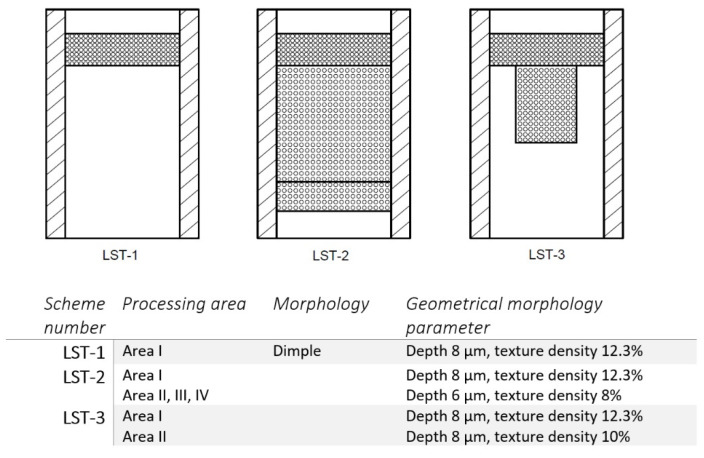
Oil pocket arrays on the surface of the cylinder liner, after [116].

**Figure 18 materials-15-08629-f018:**
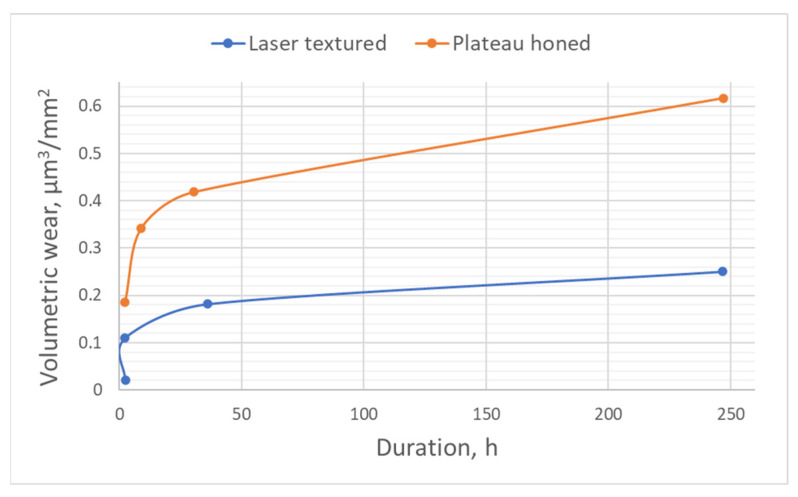
Relative volumetric wear levels of cylinder liners, after [118].

**Figure 19 materials-15-08629-f019:**
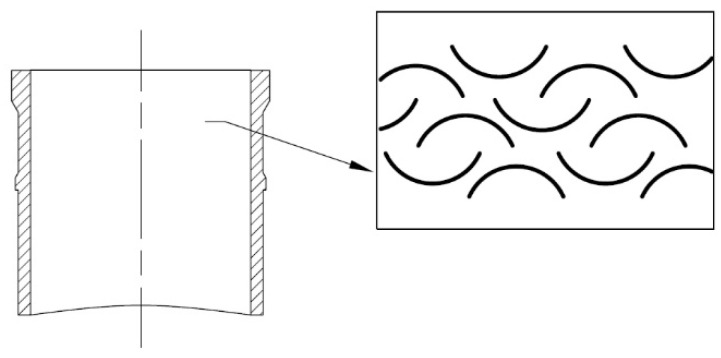
Laser-etched cylinder liner, after [120].

**Figure 20 materials-15-08629-f020:**
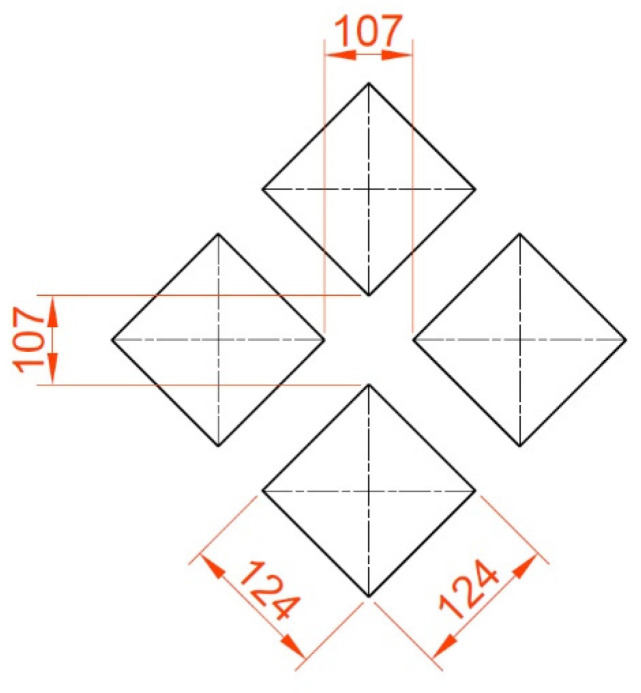
Indented oil pockets, after [122].

**Figure 21 materials-15-08629-f021:**
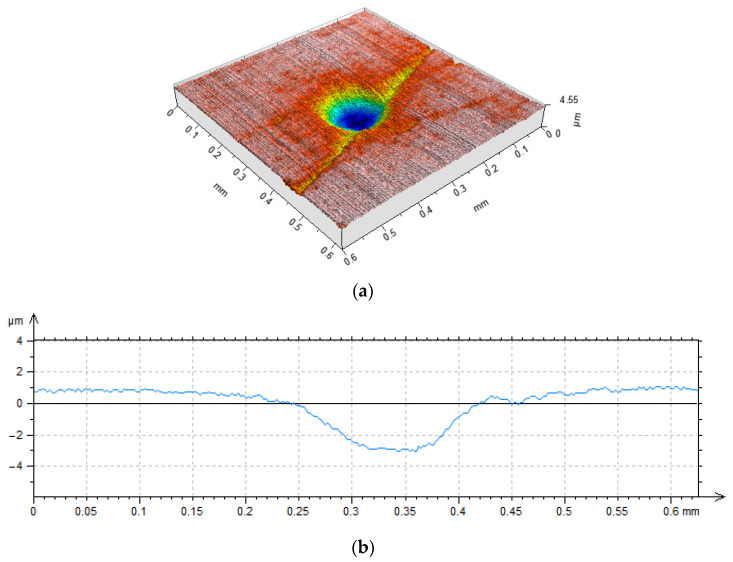
View (**a**) and profile (**b**) of the dimple of the cylinder surface used in [123].

**Figure 22 materials-15-08629-f022:**
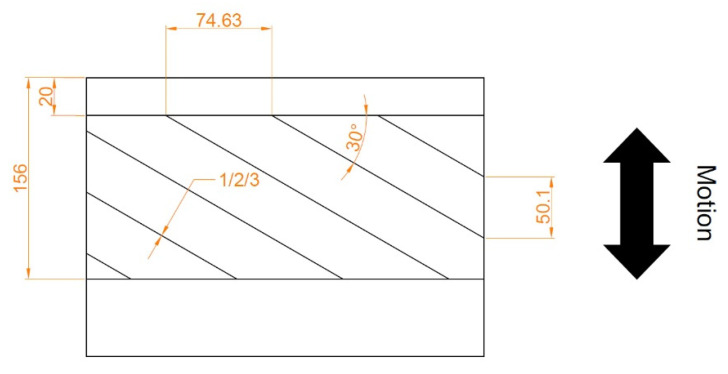
Cylinder liner tested in [29].

**Figure 23 materials-15-08629-f023:**
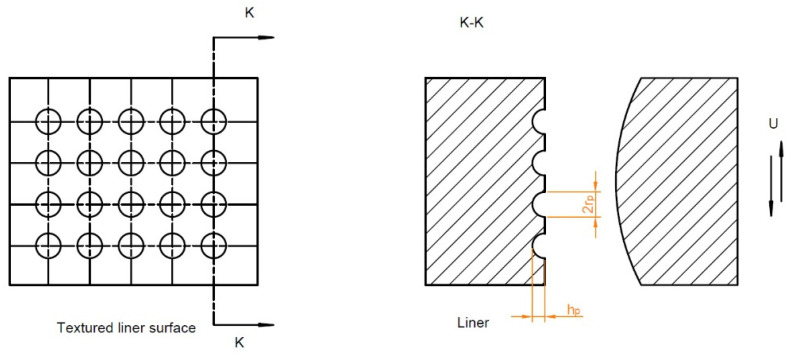
Schematic diagram of the geometry of dimples on the cylinder liner surface, after [134].

**Figure 24 materials-15-08629-f024:**
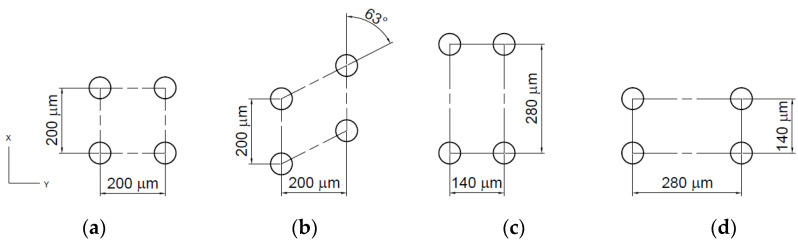
Array modes in the cylinder liner: square array (**a**), stagger array (**b**), stretching along the liner axis array (**c**), shortening along the liner axis array (**d**), after [104].

**Figure 25 materials-15-08629-f025:**
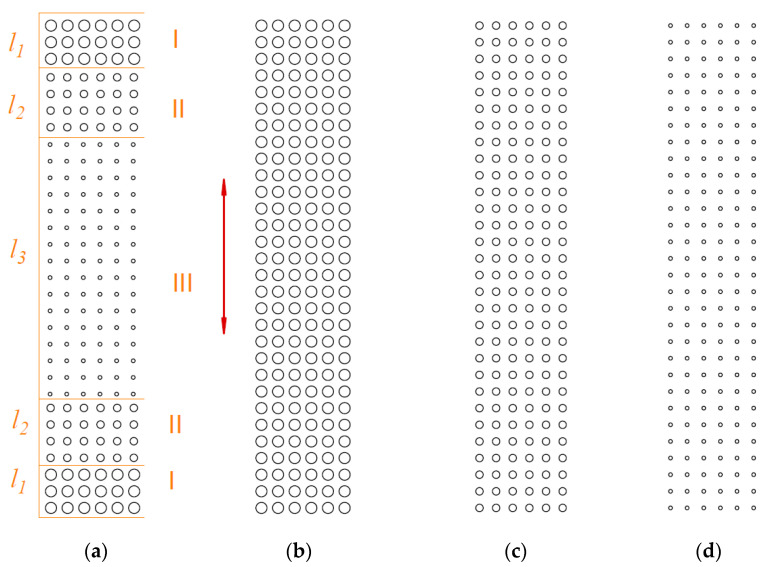
Different dimple patterns on the cylinder liner surface (**a**) surface texture with pit area ratios Sp and the dimple depth over diameter ratios e in different regions (in region I Sp = 19%, e = 0.1, in region II Sp = 16%, e = 0.09 and in region III Sp = 11%, e = 0.08 (**b**) surface texture with Sp = 19%, e = 0.1 (**c**) surface texture with Sp = 16%, e = 0.09, (**d**) surface texture with Sp = 11%, e = 0.08; l_1_ = 16,2 mm, l_2_ = 25.2 mm, l_3_ = 97.2 mm, after [135].

**Figure 26 materials-15-08629-f026:**
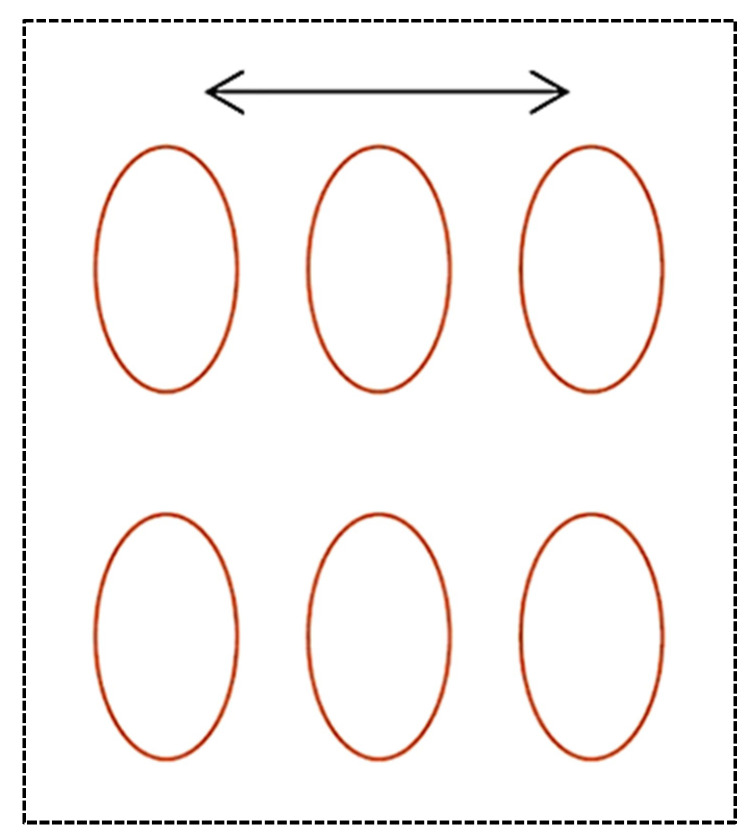
Array of dimples, after [137].

**Figure 27 materials-15-08629-f027:**
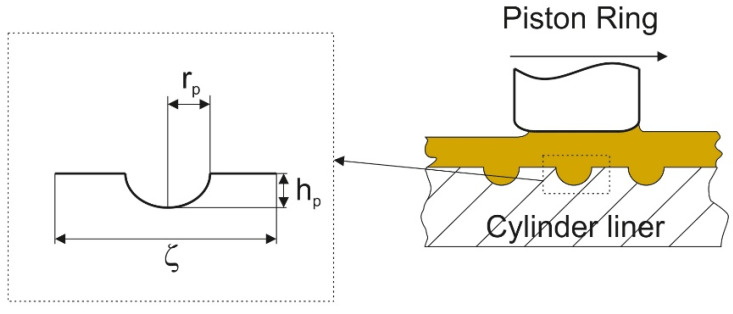
The scheme of contact between the cylinder liner and the piston ring, after [141].

**Figure 28 materials-15-08629-f028:**
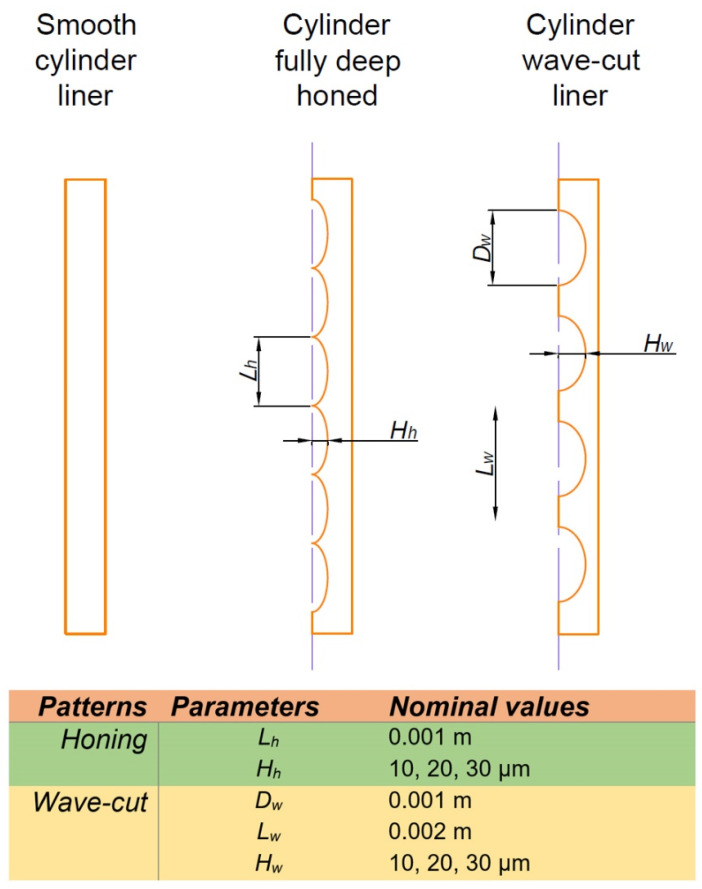
Scheme of the surface of the cylinder, after [142].

**Table 1 materials-15-08629-t001:** Research presented in References.

Type of Research		Texturing Effects		
Laboratory simulators		Friction		Wear
	Test rigs	[27,59,60,61,64,66,67,68,70,71,72,73,74,76,77,78,79,80,81,84,85,87,88,89,90,91]		[27,59,60,62,63,64,65,66,67,71,73,80,81,83,85,88]
	Engine simulators	[43,103,104,105]		[101,102]
Tests of fired engines		Oil consumption	Wear	Parameters (torque, power, fuel consumption)
		[106,115,116,117,118,119,121,124,125]	[29,118,119,124]	[105,120,121,122,123]
Modeling		Friction, oil film thickness		
		[43,64,68,77,78,104,121,128,129,130,132,133,134,135,136,137,139,140,141,142,143,144,145,146,147]		

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
