# Peer review of "Surface Texturing of Cylinder Liners: A Review"

_materials, 2022, doi:10.3390/ma15238629_

Round 1
Reviewer 1 Report
The review is written at a very high level. There are only 2 recommendations:
1. At analysing the impact of regular microreliefs on operational properties, it would be advisable to analyse the article on the formation of partially regular microreliefs, like [https://journals.vilniustech.lt /index.php/Aviation/article/view/15889/10845], where experimental studies of the influence of technological parameters during the formation of partially regular microreliefs on the end surfaces of the bodies of rotation on the geometric parameters of the formed microrelief are given.
2. The authors should also consider the effect of surface roughness on the performance of internal cylindrical surfaces, which is given in the paper [https://doi.org/10.3390/machines9060116], where it is indicated that the formation of a regular microrelief on such surfaces is mandatory to ensure the necessary oil capacity of the surface and how result - given resource of the product. The results of experimental studies carried out on the sleeve of the hydraulic cylinder of an automobile crane given in this article confirm this. This would correlate with the research of the authors of the article shown in Fig. 11. (279…286).
Author Response
The review is written at a very high level. There are only 2 recommendations:
- At analysing the impact of regular microreliefs on operational properties, it would be advisable to analyse the article on the formation of partially regular microreliefs, like [https://journals.vilniustech.lt /index.php/Aviation/article/view/15889/10845], where experimental studies of the influence of technological parameters during the formation of partially regular microreliefs on the end surfaces of the bodies of rotation on the geometric parameters of the formed microrelief are given.
- The authors should also consider the effect of surface roughness on the performance of internal cylindrical surfaces, which is given in the paper [https://doi.org/10.3390/machines9060116], where it is indicated that the formation of a regular microrelief on such surfaces is mandatory to ensure the necessary oil capacity of the surface and how result - given resource of the product. The results of experimental studies carried out on the sleeve of the hydraulic cylinder of an automobile crane given in this article confirm this. This would correlate with the research of the authors of the article shown in Fig. 11. (279…286).
These References have been added.
Reviewer 2 Report
There are (already) a number of reviews on surface texturing. This publication provides an overview of the relevant literature - mainly recent - with a focus on the application of surface texturing in the field of internal combustion engines, specifically cylinders, cylinder liners, pistons and piston rings. Yet, in addition to some spelling mistakes and typos, possibly faulty transfer from source file as well as incorrect expressions are unfortunately to be complained about, often e.g. the sloppy synonymous use of "friction" and "coefficient of friction".
A major shortcoming is that reference literature is mentioned, but not infrequently only in general terms (more or less commonplaces), while the respective study conditions and results or their explanations are not cited accordingly, not even in a summarising form that concentrates on the essentials. Especially in these evaluations, there is no (critical) discussion of the results. Regrettably, this unnecessarily limits the technical content value and the didactic effect of the publication.
The uploaded file (PDF), which has been commented on many times, points out the criticised passages in detail, showing in respective "comment fields" directly recommended text changes without further notes, comments and questions in square brackets.
The authors should be invited to make changes and especially additions in this sense. The usefulness and quality of the presented rerview could also be increased if the summary were supplemented - as an original contribution of the author team to the subject area - by suitable, possibly abstracted or simplified, overview tables and/or graphics.

Author Response
There are (already) a number of reviews on surface texturing. This publication provides an overview of the relevant literature - mainly recent - with a focus on the application of surface texturing in the field of internal combustion engines, specifically cylinders, cylinder liners, pistons and piston rings. Yet, in addition to some spelling mistakes and typos, possibly faulty transfer from source file as well as incorrect expressions are unfortunately to be complained about, often e.g. the sloppy synonymous use of "friction" and "coefficient of friction".
A major shortcoming is that reference literature is mentioned, but not infrequently only in general terms (more or less commonplaces), while the respective study conditions and results or their explanations are not cited accordingly, not even in a summarising form that concentrates on the essentials. Especially in these evaluations, there is no (critical) discussion of the results. Regrettably, this unnecessarily limits the technical content value and the didactic effect of the publication.
The uploaded file (PDF), which has been commented on many times, points out the criticised passages in detail, showing in respective "comment fields" directly recommended text changes without further notes, comments and questions in square brackets.
The authors should be invited to make changes and especially additions in this sense. The usefulness and quality of the presented review could also be increased if the summary were supplemented - as an original contribution of the author team to the subject area - by suitable, possibly abstracted or simplified, overview tables and/or graphics.
Thank you for accurate review. We have made changes suggested in the uploaded file. Critical discussion of the results is given in point 3 with some shortcomings such as not considering oil capacity by researchers, small number of works focused on both modeling and experiment, analysis of asperity contact using old models, small number of experimental tests under elevated temperatures. We have added table 1, summarizing research presented in References.
Round 2
Reviewer 2 Report
Suggested changes and additions have be performed. The paper reached appropriate quality. Yet I found stil some mistakes to be corrected.
Page 3, line 125: I guess film thickness is near 1 µm!
Page 4, Fig. 3: Not necessary so many digits (indication of integers is enough)! The conditions of evaluation - test duration, ... - should be given, e.g. in the caption!
Page 10, Fig. 10: Just for clarity: Indicate as well sliding direction!
Page 24, line 658: ... was thicker for laser-patterned
Author Response
Suggested changes and additions have be performed. The paper reached appropriate quality. Yet I found still some mistakes to be corrected.
- Page 3, line 125: I guess film thickness is near 1 µm!
Improved.
- Page 4, Fig. 3: Not necessary so many digits (indication of integers is enough)! The conditions of evaluation - test duration, ... - should be given, e.g. in the caption!
Improved. Information on test conditions has been added.
- Page 10, Fig. 10: Just for clarity: Indicate as well sliding direction!
Done.
- Page 24, line 658: ... was thicker for laser-patterned.
Improved.